# Antifungal Activity of Plant Waste Extracts against Phytopathogenic Fungi: *Allium sativum* Peels Extract as a Promising Product Targeting the Fungal Plasma Membrane and Cell Wall

Ana Teixeira [1,2], Eva Sánchez-Hernández [3], João Noversa [1,4], Ana Cunha [1,4], Isabel Cortez [2,5], Guilhermina Marques [2,5], Pablo Martín-Ramos [3] and Rui Oliveira [1,4,*]

1   Department of Biology, School of Sciences, University of Minho, Campus de Gualtar, 4710-057 Braga, Portugal
2   Centre for Research and Technology of Agro-Environmental and Biological Sciences (CITAB), Inov4Agro, University of Trás-os-Montes and Alto Douro (UTAD), Quinta de Prados, 5000-801 Vila Real, Portugal
3   Department of Agricultural and Forestry Engineering, ETSIIAA, Universidad de Valladolid, Avenida de Madrid 44, 34004 Palencia, Spain
4   Centre of Molecular and Environmental Biology (CBMA), University of Minho, Campus de Gualtar, 4710-057 Braga, Portugal
5   Department of Agronomy, School of Agrarian and Veterinary Sciences, University of Trás-os-Montes and Alto Douro (UTAD), Quinta de Prados, 5000-801 Vila Real, Portugal
*   Correspondence: ruipso@bio.uminho.pt

**Abstract:** The harmful effect of synthetic fungicides on the environment and the development of resistance by fungi raises concerns about their security and future efficacy. In this work, we investigated plant by-products with the antifungal activity that could be safe alternatives to conventional fungicides. The in vitro antifungal potential of plant by-product extracts showed that garlic peel extract (GPE) was the most effective against several phytopathogenic fungi. Accordingly, in ex situ assays with apples, GPE significantly reduced the lesion size caused by subepidermal inoculation with *Colletotrichum acutatum* spores. In addition, *Saccharomyces cerevisiae* mutant strains affected in ergosterol synthesis showed higher resistance to GPE than the parental strain, indicating that the extract might target an intermediate of this pathway. Moreover, GPE affects the cell wall, given that *bck1* and *mkk1/mkk2* mutants were less able to cope with the stress because of the impairment of the remodeling mechanisms. Regarding the apoptosis-deficient mutant *yca1*, sensitivity was similar to that of the parental strain, suggesting that the extract does not induce apoptosis. A diverse group of sulfur compounds was identified by gas chromatography–mass spectrometry (GC/MS). Our findings contribute to the elucidation of the antifungal mechanism of GPE and highlight its potential as an alternative biofungicide in agriculture.

**Keywords:** plant residues; antifungal activity; biofungicide; garlic peels; antifungal action mechanism; ergosterol biosynthesis; fungal cell wall damage

## 1. Introduction

According to the Food and Agriculture Organization (FAO), over the next 35 years, to satisfy the food demand, the world's food production should increase by over 60% [1]. However, food security faces a number of challenges across both production and consumption. With the growing world population, by 2050, it is projected that we will need 120% more water, 42% more cropland, lose 14% more forest, and produce 77% more greenhouse gas emissions [2]. Paradoxically, only in European Union, in 2016, 2.6 million tons of food waste were produced and cast-off to landfills [3]. In this scenario, improvement of food security is pivotal to ensuring good quality nutrition for the population with minimal impact on the environment.

Physiological and environmental factors (availability of essential inorganic nutrients, drought, and salinity) and the physical and chemical nature of the soil can negatively affect plant growth [4]. Additionally, pests and diseases caused by viruses, bacteria, fungi, and insects tend to increase, leading to crop and livestock damage with huge impacts on food production, animal welfare, and economic losses [5,6]. The large-scale application of fertilizers and pesticides was vital for stabilizing crop yield to guarantee high productivity and food security. However, the negative impacts of these agrochemicals on natural habitats, the environment, and human health are dramatic [7–10]. Beyond this, the effectiveness of pesticides is threatened by the evolution of resistance in target and non-target (micro)organisms [11–13]. Thus, the rapid development of resistance to agrochemicals and consequent control failure has become increasingly problematic.

Sustainability can be increased in crop production by expanding the use of bioactive natural products for plant protection [14]. Plant compounds and their derivatives are still one of the major sources of new drug molecules today, not only for treating human diseases but also for agricultural applications, i.e., as biopesticides [15,16]. Plants are rich in a wide variety of secondary metabolites of ecological significance, such as tannins, terpenoids, alkaloids, and flavonoids. Some of these metabolites can act as biopesticides and might be useful for disease control in agriculture [17]. For instance, quaternary benzophenanthridine alkaloids, isolated from *Macleaya cordata* leaf, are shown to be effective against powdery mildew of wheat caused by *Erysiphe graminis* and tomato grey mold caused by *Botrytis cinerea* [18], and the coumarins isolated from citrus peels demonstrated antifungal activity against the causing agent of anthracnose disease, *Colletotrichum* sp. [19]. Furthermore, saponins isolated from *Chenopodium quinoa* seeds have high efficacy for the treatment of pea root rot and tomato early blight [20], and the terpene geraniol found in lemon thyme (*Thymus citriodorus*) showed great nematicidal potential against two plant parasitic nematodes (*Meloidogyne incognita* and *Meloidogyne javanica*) [21]. Accordingly, phytopesticides are receiving increasing attention as an alternative to synthetic pesticides for the management of plant diseases due to their advantages in terms of safety, easy biodegradability, environmental friendliness, and low toxicity [16]. Furthermore, the regulation of the European Union (Article 14 in Directive 2009/128/EC, Council Regulation (EC) 834/2007, Commission Regulation (EC) 889/2008, Regulation (EU) 2019/1009) promotes the use of formulations based on natural products for integrated pest management.

In order to counter the global risks to plant health and meet the goal of food security in a changing environment, a more holistic approach is required, integrating plant breeding, ecology, biotechnology, and sustainable management practices. In this work, we intended to identify and valorize new sources of plant metabolites with antifungal activity from plant by-products. These plant wastes normally end up as residues in landfills, so their valorization as a source of antifungal compounds would channel them back into the production chain and comply with a circular economy. Therefore, the effectiveness of different plant by-product matrices against selected phytopathogenic fungi was evaluated, both in vitro and ex situ assay, and the action mechanism of the most promising plant waste was investigated.

## 2. Materials and Methods

### 2.1. Microorganisms, Culture Media and Growth Conditions

The fungi—*Diplodia corticola* (provided by Ana Cristina Esteves, Centre for Environmental and Marine Studies, CESAM, University of Aveiro), *Phytophthora cinnamomi* (provided by Helena Machado of the National Institute for Agrarian and Veterinarian Research, INIAV, Lisbon), *B. cinerea* (provided by Richard Breia, Centre of Molecular and Environmental Biology, CBMA, University of Minho), and *C. acutatum* and *Colletotrichum nymphaeae* (provided by Pedro Talhinhas, School of Agriculture, University of Lisbon), were cultured in potato dextrose agar (PDA) medium (Biolife) at 25 °C in the dark. The yeast strains used were *Saccharomyces cerevisiae* BY4741 (MATa; *his3Δ1; leu2Δ0; met15Δ0; ura3Δ0*) from Euroscarf, the ergosterol synthesis mutants *erg2* (BY4741; MATa; *erg2Δ::kanMX*), *erg4*

(BY4741; *MATa*; *erg4Δ::kanMX*), *erg5* (BY4741; MAT*a*; *erg5Δ::kanMX*) and *erg6* (BY4741; MAT*a*; *erg6Δ::loxP-kanMX-loxP*) provided by Marie Kodedová and Hana Sychrová from the Czech Academy of Sciences, Prague, Czech Republic; the cell wall defective *bck1* (BY4741; MAT*a*; *his3Δ1; leu2Δ0; met15Δ0; ura3Δ0; YJL095w::kanMX4*) and *mkk1/mkk2* (BY4741; MAT*a*; *his31; leu20; met150; ura30; mkk2::kanMX4; mkk1::LEU2*) strains. The parental strain W303-1A (MAT*a*; *ade2-1; ura3-1; leu2- 3,112; trp1-1; his3-11,15; can1-100*), as well as the mutant *yca1* (W303-1A; MAT*a*; *yca1::kanMX4*) were provided by Francesc Posas from Universitat Pompeu Fabra, Barcelona, Spain. The yeast strains were cultivated on YPD solid medium [1% *w/v* yeast extract (Acros Organics), 2% *w/v* peptone (Biolife); 2% *w/v* dextrose (Scharlau); 2% *w/v* agar (LabChem)] at 30 °C for two days, and then stored at 4 °C.

## 2.2. Plant By-Products and Extracts Preparation

Eleven plant by-products were used in this work and were obtained from May to September 2021. Peels from banana, garlic, lemon, orange, brown onion, pomegranate, and white potatoes were purchased in local stores; barks from eucalyptus and pine, pine needles, and olive leaves were collected in Marco de Canaveses, Porto, Portugal. Peels from banana, lemon, orange, pomegranate, and potato were removed manually, washed with deionized water, and kept at −80 °C until the lyophilization process. Garlic and onion peels, eucalyptus and pine bark, pine needles, and olive leaves were rinsed with deionized water and dried at room temperature (rt), protected from light. The residues were ground into a fine powder using an electric blender, and 10 g of biomass of each plant material was extracted in 80% (*v/v*) ethanol in a ratio of 1 g:10 mL with stirring cycles at rt in the dark for 48 h. The extracts were then centrifuged (5000 rpm, 8 min; Eppendorf 5804R), filtrated (Whatman filter paper 1), concentrated using a rotary evaporator (40 °C, 100 rpm) and stored at −80 °C. After lyophilization, extracts were kept in the dark and protected from moisture.

## 2.3. Screening of Extracts Antifungal In Vitro Activity

The effect of plant by-product extracts on the mycelial growth of phytopathogenic fungi was evaluated through the Poisoned Method Food [22]. Briefly, the ethanolic extracts were incorporated into a PDA medium at 750 µg/mL and poured into Petri dishes. Five mm diameter mycelial discs of *D. corticola*, *B. cinerea*, *C. nymphaeae*, and *P. cinnamomi* were removed from the margins of 10-day-old cultures and placed in the middle of the prepared plates. The plates were incubated at 25 °C, protected from light, and the colony diameter of the fungi was measured periodically. The experiment ended when the negative control reached full growth in the Petri dish. The negative control consisted in replacing the extract with 50% (*v/v*) ethanol (solvent used to resuspend the extracts) in the PDA medium. The inhibitory activity of each extract was calculated as the percent growth inhibition compared to the negative control (0% of inhibition), according to Formula (1):

$$\text{Inhibition (\%)} = \frac{dc - de}{dc} \times 100 \tag{1}$$

where *dc* is the diameter of control mycelium, and *de* is the diameter of mycelium exposed to extracts.

## 2.4. Evaluation of Extract Antifungal Using an Ex Situ Apple Assay

The ex situ antifungal activity of garlic peel extract (GPE) was assessed in 'Golden' cultivar apples according to Pereira et al. [23] and Loebler et al. [24]. The fruits were cultivated according to organic farming guidelines, collected from the orchard in October 2022, and used immediately. Briefly, apples without visible lesions were chosen for the experiment and disinfected by dipping in 0.02% (*v/v*) sodium hypochlorite for 10 min, washed in tap water three times, and left to dry out. Under aseptic conditions, each fruit was perforated with a truncated needle in two equidistant points (3 mm diameter × 15 mm depth) and filled with 20 µL of 25, 50, or 100 mg/mL GPE or 50% (*v/v*) ethanol. One hour later, wounds

were inoculated with 8 μL of *C. acutatum* spore suspension ($5 \times 10^4$ spores/mL) or sterile deionized water for control of the injection procedure. The spores were removed from a three-week-old *C. acutatum* PDA culture by shaking the Petri dish horizontally with 15 mL sterile deionized water covering the mycelium. The suspension was collected with a micropipette and filtered through four layers of sterile cheesecloth in order to remove adhered mycelium and agar. Spore concentration was determined using a Neubauer chamber and further adjusted with sterile deionized water. Apples were then placed in clean boxes (one box per treatment), separated from each other, wrapped with plastic film, and maintained at RT for 12 days. Lesion diameters (LD) were measured daily, and the lesion size reduction percentage (% LSR) in comparison with the positive control (0% reduction) was determined according to Formula (2):

$$\text{LSR } (\%) = \frac{LSc - LSt}{LSc} \times 100 \tag{2}$$

where *LSc* is the lesion diameter of the positive control, and *LSt* is the lesion diameter of treated fruits. At the end of the experiment (day 12), apples were cut through the middle of the lesions in the cuticle, and the inner lesions were photographed.

### 2.5. Cell Viability Assays

Exponentially growing cultures of *S. cerevisiae* (described in Section 2.1) were treated with 750 μg/mL GPE, or 50% (*v/v*) ethanol (the same volume as the GPE treatment) as a negative control, followed by incubation at 30 °C and 200 rpm. Aliquots of 100 μL were taken at 0, 1, and 2 h of treatment, serially diluted in sterile deionized water to $10^{-4}$, and subsequently, aliquots of 40 μL were spread on YPD solid medium. The plates were then incubated for two days at 30 °C, and the colonies were counted. Cell viability was calculated as a percentage of the colony-forming units (CFUs), taking time 0 min as 100%.

### 2.6. Chemical Characterization

The infrared spectrum of the freeze-dried extract was collected using a Nicolet iS50 (ThermoFisher Scientific; Waltham, MA, USA) Fourier-transform infrared (FTIR) spectrometer equipped with a diamond attenuated total reflection (ATR) module. Operative conditions: room temperature, 400–4000 cm$^{-1}$ range; 0.5 cm$^{-1}$ spectral resolution; coaddition of 64 scans. The extract was analysed by gas chromatography–mass spectrometry (GC–MS) at the Research Support Services (STI) at the University of Alicante (Alicante, Spain), using a gas chromatograph model 7890A coupled to a quadrupole mass spectrometer model 5975C (both from Agilent Technologies, Santa Clara, CA, USA). The operating conditions were: injector temperature = 280 °C, splitless mode; injection volume = 1 μL; initial temperature = 60 °C, 2 min, followed by a ramp of 10 °C·min$^{-1}$ to a final temperature of 300 °C, 15 min. The chromatographic column used for the separation of the compounds was an Agilent Technologies HP-5MS UI of 30 m in length, with 0.250 mm diameter and 0.25 μm film. The mass spectrometer conditions were: mass spectrometer electron impact source temperature = 230 °C and quadrupole temperature = 150 °C; ionization energy = 70 eV. Test mixture 2 for apolar capillary columns according to Grob (Supelco 86501) and PFTBA tuning standards were used for calibration, purchased from Sigma Aldrich Química S.A. (Madrid, Spain). Comparison of mass spectra and retention times with those of reference compounds and computer matching with the databases of the National Institute of Standards and Technology (NIST11).

### 2.7. Statistical Analysis

Results are expressed as mean ± standard deviation (SD) of at least three independent replicas per assay except for the ex situ assay, where four replicates were included per condition. Statistical analysis was performed using GraphPad Prism Software v8 (GraphPad Software, California, USA). One-way ANOVA followed by Tukey's post-hoc test was performed. Differences between mean values were considered statistically significant when

$p < 0.05$ (*), $p < 0.01$ (**), $p < 0.001$ (***), or $p < 0.0001$ (****). The statistical significance of the mean value comparison was also represented by lowercase letters (a and b; $p < 0.05$).

## 3. Results

### 3.1. Screening of In Vitro Antifungal Activity of Plant By-Products

The antifungal activity of eleven plant by-products was determined in vitro by mycelium growth evaluation of *D. corticola*, *B. cinerea*, *C. nymphaeae*, and *P. cinnamomi* on PDA medium (Table 1). At the end of six days, GPE was the most effective in reducing the mycelium growth of *D. corticola*, *C. nymphaeae,* and *P. cinnamomi*, showing inhibition of 68.7, 53.4, and 68.0%, respectively, followed by onion peel extract (OPE), with growth inhibitions of 42.0, 50.7, and 50.6%, respectively. In turn, the extracts of banana and pomegranate peels were the least effective at the tested concentration. In addition, our results showed that the tested phytopathogenic organisms have different sensitivities to the studied extracts at 750 µg/mL: the fungus *C. nymphaeae* and the oomycete *P. cinnamomi* were generally the most resistant organisms. Due to the stronger antifungal activity, GPE was chosen for further studies regarding ex situ assays, investigation of the mechanism of action, and chemical analysis.

**Table 1.** Inhibition of growth of *Diplodia corticola*, *Botrytis cinerea*, *Colletotrichum nymphaeae*, and *Phytophtora cinnamomi* by plant by-product extracts at 750 µg/mL after six days of incubation. The results are representative of three independent experiments and are expressed as follows: no activity (–) corresponds to 0–5% of mycelium inhibition; low activity (+) corresponds to 6–20% of mycelium inhibition; medium activity (++) corresponds to 21–49% of mycelium inhibition, and high activity (+++) corresponds to ≥ 50% of mycelium inhibition. Results are representative of mean ± SD (n = 3). NT—not tested.

| Plant by-Products Extracts (750 µg/mL) | Phytopathogenic Fungi | | | |
|---|---|---|---|---|
| | *D. corticola* | *B. cinerea* | *C. nymphaeae* | *P. cinnamomi* |
| Banana peels | + | ++ | - | - |
| Eucalyptus bark | + | + | + | - |
| Garlic peels | +++ | ++ | +++ | +++ |
| Lemon peels | ++ | ++ | - | ++ |
| Olive leaves | + | +++ | - | - |
| Onion peels | ++ | ++ | +++ | +++ |
| Orange peels | ++ | ++ | - | ++ |
| Pine bark | ++ | + | + | ++ |
| Pine needles | ++ | +++ | + | - |
| Pomegranate peels | - | ++ | - | - |
| Potato peels | ++ | ++ | - | NT |

### 3.2. GPE as a Biocontrol Agent for C. acutatum Proliferation

In order to assess the potential of antifungal activity to protect plants, the ability of GPE to control infection in 'Golden' cultivar apples inoculated with *C. acutatum* was investigated. The infection of *Colletotrichum* sp. (mainly *C. acutatum* and *C. gloeosporioides* species complexes) on apples (*Malus domestica* Borkh) can result in the entire crop loss resulting in huge economic losses in the sector [25]. The inoculation of healthy apples with *C. acutatum* induced rot of the infected zone (Figure 1A), as well as the mycelium growth around the orifice of inoculation (not shown), whereas these symptoms were reduced as GPE concentration increased, with maximum effectiveness at 100 mg/mL. The GPE showed a significant, dose-dependent external lesion diameter reduction in GPE-treated apples during all 12 days of the experiment (Figure 1B). Maximum lesion size reduction (LSR) was reached on the fifth day, with 25, 78, and 100% for 25, 50, and 100 mg/mL GPE, respectively (Figure 1C). Remarkably, there is evidence indicating that LSR did not decrease over time since, in almost all time points, this value did not change for the same treatment. Nonetheless, at the end of the experiment, the apple treatment with 100 mg/mL

GPE reduced the propagation of *C. acutatum* by 77% (Figure 1C). These results suggest that GPE could have an important role in the biocontrol of phytopathogenic organisms, such as *Colletotrichum* species, allowing food security and avoiding huge economic losses in this sector.

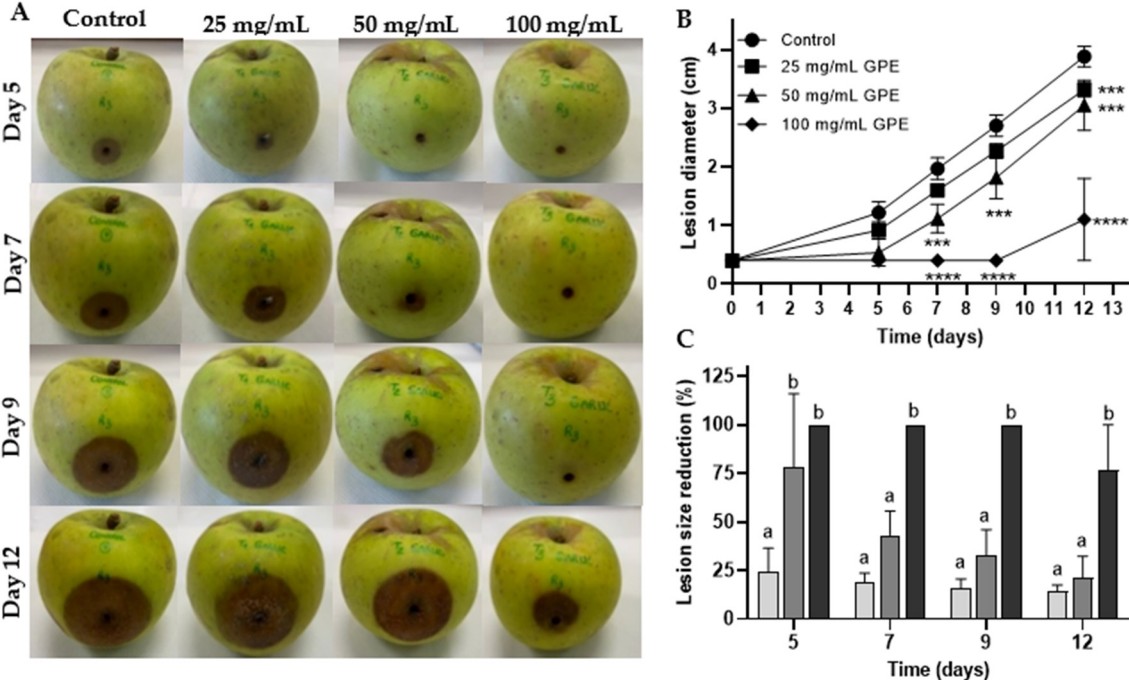

**Figure 1.** Effect of garlic peel extract (GPE) at 25, 50, and 100 mg/mL on 'Golden' cultivar apples infected with *Colletotrichum acutatum* upon inoculation. Apples were inoculated with $5 \times 10^4$ spores/mL in a manually-made orifice, and then 25, 50, or 100 mg/mL GPE were placed in the same orifice. After 12 days, apples were photographed. The control was made by replacing GPE with the solvent (50% *v/v* ethanol). Results are presented by representative photos depicting external lesions caused by the phytopathogenic fungus (**A**), by the external lesion size diameter (**B**), and by the percentage of external lesion size reduction (**C**). The negative control (solvent of the extract, 50% *v/v* ethanol) is represented by black circles (**B**), the treatment with 25 mg/mL by squares (**B**) and light grey columns (**C**), the treatment with 50 mg/mL by triangles (**B**), and medium grey columns (**C**), and 100 mg/mL by diamonds (**B**), and dark grey columns (**C**). Results are representative of 4 replicates, and results in (**B**) and (**C**) are presented as mean ± SD (n = 4). The asterisk notation [*p* < 0.001 (\*\*\*) or *p* < 0.0001 (\*\*\*\*)] indicates the statistically significant differences obtained for days 7, 9, and 12 concerning the external lesion diameters (**B**), and the lowercase letter notation indicates for each sampling time (days 5, 7, 9 and 12), that means with different letters are statistically different (**C**).

The internal lesions caused by the infection of *C. acutatum* in 'Golden' apples were also assessed to investigate the capacity of GPE to protect the fruit in the mesocarp and endocarp. As expected, the solvent of GPE did not significantly affect apples as the lesions were considerably smaller than in inoculated apples (Figure 2A,E,I). In addition, lesions in apples treated only with GPE showed similar lesion sizes as the apples with only the solvent, regardless of GPE concentration (Figure 2A–D), suggesting that GPE does not affect this fruit. Although differences between 25 mg/mL (Figure 2F,J) and 50 mg/mL GPE (Figure 2G,K) were not totally evident when compared with the control (Figure 2E,I), the treatment with 100 mg/mL GPE resulted in a notorious decrease not only in the lesion area but also in the lesion depth (Figure 2H,L). Additionally, the internal lesion of the positive control group was considerably more intense in colour than those of the fruits treated with the extract, suggesting that GPE was effective in controlling the fungal proliferation after 12 days.

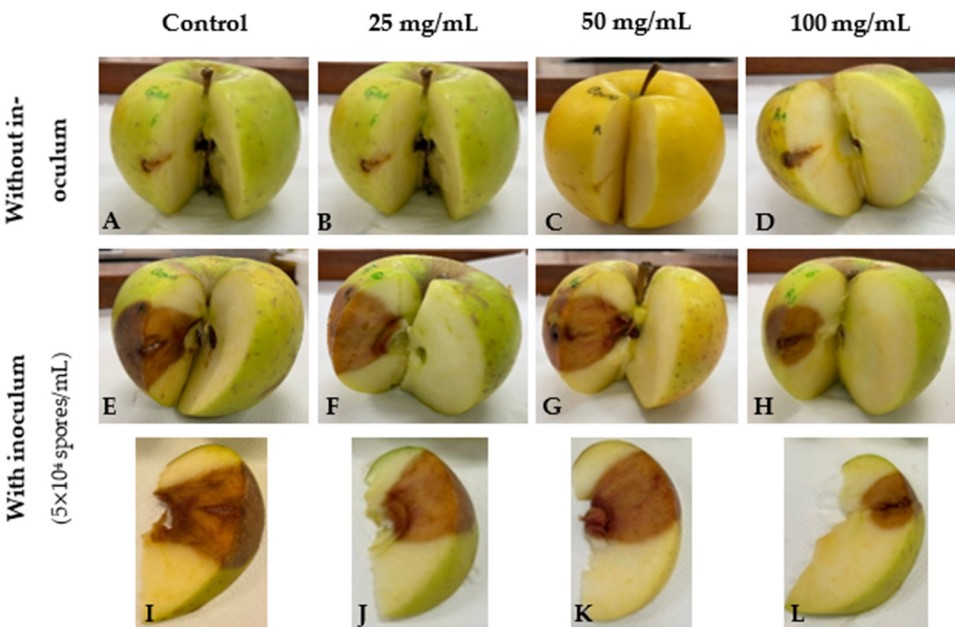

**Figure 2.** Internal lesions caused by *Colletotrichum acutatum* in 'Golden' cultivar apples upon inoculation with *C. acutatum* spores in the presence and absence of garlic peel extract (GPE). Apples were inoculated with $5 \times 10^4$ spores/mL in a manually-made orifice, and then 25 mg/mL (**F,J**), 50 mg/mL (**G,K**), or 100 mg/mL GPE (**H,L**) was placed in the same orifice. Punctured apples without inoculum and with 25 mg/mL (**B**), 50 mg/mL (**C**), or 100 mg/mL GPE (**D**) were also prepared. After 12 days, apples were cut through the middle of the orifice to remove a slice and photographed (**A–H**). The slice was also photographed to show the internal lesions more clearly (**I–L**). The control (**A,E,I**) was prepared by replacing GPE with the solvent (50% *v/v* ethanol). Results are representative of 4 replicates.

### 3.3. Toxicity Mechanism of GPE in S. cerevisiae Fungal Model

*Saccharomyces cerevisiae* has been used as a model of fungi in studies regarding the inhibitory effect of various compounds against phytopathogenic fungi [26,27]. Thus, to understand the underlying GPE action mechanism associated with its toxicity on fungi, several common antifungal toxicity targets, such as the ergosterol biosynthesis pathway, cell death, and cell wall, were investigated. Regarding ergosterol biosynthesis, we investigated the effect of GPE in mutant strains affected by gene-encoding enzymes from the final steps of this pathway (*ERG6*, *ERG2*, *ERG5*, and *ERG4*). The disruption of *ERG6* blocks the conversion of zymosterol into fecosterol by sterol C-24-methyltransferase, resulting in the accumulation of the former. *ERG2* is responsible for converting fecosterol into episterol precursor by sterol C-8 isomerase. The mutant strains *erg5* and *erg4* do not have functional *ERG5* and *ERG4* genes, which encode enzymes that catalyze two final reactions of the pathway, namely the conversion of ergosta-5,7,42(28)-trienol into ergosta-5,7,22,24(28)-tetraenol and further into ergosterol, respectively [28]. So, if GPE interferes with ergosterol biosynthesis, it is expected that at least some of the *erg* mutant strains might be more resistant to the extract in comparison to the parental strain since ergosterol and/or some of its precursors might be missing as targets of components of the extract. On the other hand, if GPE targets the plasma membrane by destabilization, it is expected that all mutant strains would be more sensitive due to the lack of ergosterol in this organelle. The comparative analysis of the parental strain and *erg* mutants showed that *erg6*, *erg5*, and *erg4* mutant strains were more resistant to GPE, showing statistically significant differences after 2 h of incubation (Figure 3A,C,D). However, unlike these mutants, the *erg2* mutant displayed similar viability as the parental strain when exposed to 750 µg/mL GPE (Figure 3B). The fact that the mutants did not display higher sensitivity than the parental strain suggests that GPE does not target the plasma membrane by destabilization. Regarding the higher

resistance of *erg6*, *erg5*, and *erg4* in relation to the parental strain, the lack of specific targets of the ergosterol biosynthesis pathway might be the reason for this behaviour. Given that fecosterol accumulates when *ERG2* is deleted, unlike deletions in *ERG6*, *ERG5*, and *ERG4*, this precursor might be targeted by one or more components of GPE.

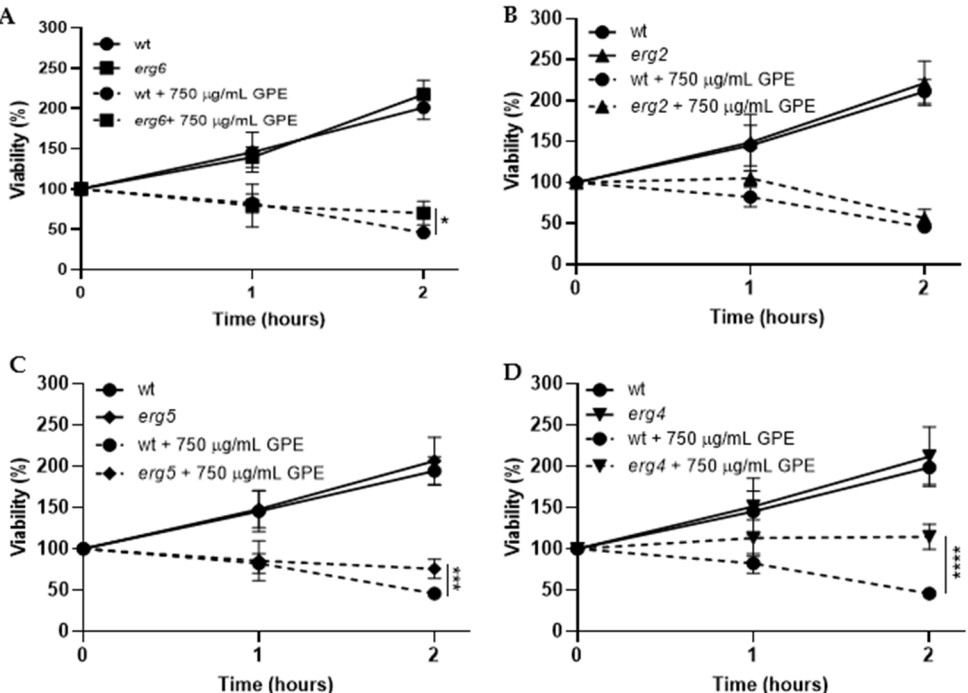

**Figure 3.** Viability of *Saccharomyces cerevisiae* parental strain BY4741 (circles) and of the derived mutant strains affected in the ergosterol biosynthesis pathway *erg6* (**A**; squares), *erg2* (**B**; triangles), *erg5* (**C**; diamonds), and *erg4* (**D**; inverted triangles), upon 2 h exposure to 750 µg/mL garlic peel extract (GPE). Cells from exponentially growing cultures were incubated with GPE (dashed lines) or with the same volume of the solvent (50% *v/v* ethanol; solid lines) at 30 °C, 200 rpm, and aliquots were harvested along time, serially diluted to $10^{-4}$, and spread on YPD solid medium. Upon incubation at 30 °C for 48 h, colonies were counted, and viability was calculated, taking time of 0 min as 100% viability. Values represent the mean ± SD of three independent experiments. One-way ANOVA followed by Tukey's post-hoc test was used; statistical differences are pointed at 2 h of incubation with * if $p < 0.05$, *** if $p < 0.001$, and **** if $p < 0.0001$.

Apoptosis is a programmed cell death process that plays a crucial role in the maintenance of cellular homeostasis by removing damaged or dispensable cells in multicellular eukaryotic organisms [29,30]. The *yca1* mutant strain does not possess the *YCA1* gene that encodes the protein metacaspase Yca1, reported as one of the positive regulators of apoptosis in *S. cerevisiae* when activated upon stress [31–33]. Thus, the disruption of the *YCA1* gene increases cellular resistance to stress by blocking programmed cell death [34]. According to the results depicted in Figure 4, the *yca1* mutant strain displayed similar resistance as the parental strain when exposed to GPE, suggesting that the toxicity activity of the extract does not require the presence of the functional Yca1 protein and, consequently, does not induce apoptosis.

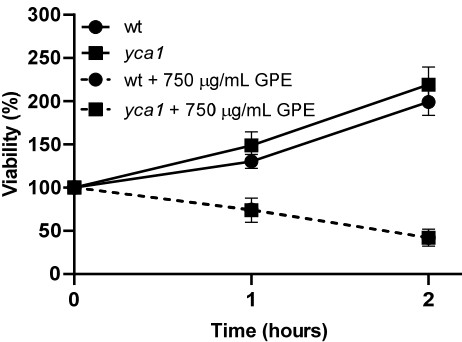

**Figure 4.** Viability of *Saccharomyces cerevisiae* parental strain W303-1A (circles) and the derived mutant strain *yca1* (squares) upon 2 h exposure to 750 µg/mL garlic peel extract (GPE). Cells from exponentially growing cultures were incubated with GPE (dashed lines) or with the same volume of the solvent (50% *v/v* ethanol; solid lines) at 30 °C, 200 rpm, and aliquots were harvested along time, serially diluted to $10^{-4}$, and spread on YPD solid medium. Upon incubation at 30 °C for 48 h, colonies were counted, and viability was calculated, taking time of 0 min as 100% viability. Values represent the mean ± SD of three independent experiments. One-way ANOVA followed by Tukey's post-hoc test was used.

Cell wall integrity, fundamental for fungal cell viability, is regulated by phosphorylation of the protein kinase C, Pkc1, which activates a signal transduction cascade with successive phosphorylations of Bck1, the redundant pair, Mkk1 and Mkk2, and the Slt2 protein. Upon activation by cell wall damage, this signalling pathway generates a response of activation of transcription of genes encoding enzymes that remodel the cell wall [35]. So, to investigate whether or not GPE targets the cell wall, we used the *bck1* and *mkk1/mkk2* mutants and challenged them with GPE. If the extract interferes with the cell wall, the mutants *bck1* and *mkk1/mkk2* would be more sensitive in comparison to the parental strain, resulting in decreased viability. As depicted in Figure 5, the exposure of the mutants *bck1* (Figure 5A) and *mkk1/mkk2* (Figure 5B) to 750 µg/mL GPE resulted in a statistically significant decrease of cell viability in relation to the parental strain. These results suggest that GPE affects the cell wall, which, in the case of the *bck1* and *mkk1/mkk2* mutants, would be less able to cope with the stress because of the impairment of the remodelling mechanisms.

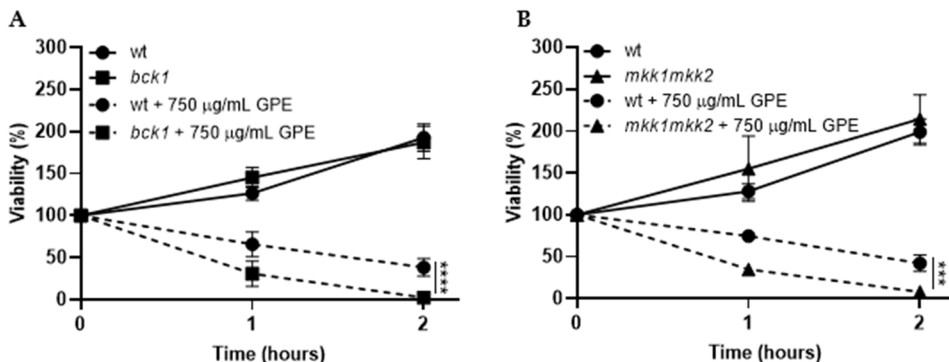

**Figure 5.** Viability of *Saccharomyces cerevisiae* parental strain BY4741 (circles) and the derived mutant strains affected in cell wall remodelling *bck1* (**A**; squares) and *mkk1/mkk2* (**B**; triangles) upon 2 h exposure to 750 µg/mL garlic peel extract (GPE). Cells from exponentially growing cultures were incubated with GPE (dashed lines) or with the same volume of the solvent (50% *v/v* ethanol; solid lines) at 30 °C, 200 rpm, and aliquots were harvested along time, serially diluted to $10^{-4}$, and spread on YPD solid medium. Upon incubation at 30 °C for 48 h, colonies were counted, and viability was calculated, taking time of 0 min as 100% viability. Values represent the mean ± SD of three independent experiments. One-way ANOVA followed by Tukey's post-hoc test was used; statistical differences are pointed at 2 h of incubation with *** if $p < 0.001$ and **** if $p < 0.0001$.

*3.4. Chemical Characterization of GPE*

3.4.1. Vibrational Characterization by ATR-FTIR

The garlic peel extract spectrum (Figure S1) exhibits bands at 3278 cm$^{-1}$ (CH=CH2, allyl; N-H stretching, amide), 2925 cm$^{-1}$, and 2856 cm$^{-1}$ (doublet, NH2) 1613 cm$^{-1}$ (C=C, allyl), 1519 cm$^{-1}$ (C=S, N-H bending, and C-N str., amide II), 1452 cm$^{-1}$ (C=S); 1401 cm$^{-1}$ (C=O str. (sym.), COO- ), 1346 cm$^{-1}$ (C=S str.); 1264 cm$^{-1}$ and 1225 cm$^{-1}$ (P=O str. (sym.), phosphodiesters), 1124 cm$^{-1}$ (C-N str., amino acids), 1109 cm$^{-1}$ (C=S str.), 1023 cm$^{-1}$ (S=O str (sym.), 971 cm$^{-1}$ (C-N str. alcohols, carboxylic acids, ethers, and esters), 927 cm$^{-1}$ (C-S-group; N-H str., proteins), 871 cm$^{-1}$ (-CC- str.; C-H bending, glycogen), 775 cm$^{-1}$ (C-H bending, glycogen), 522 cm$^{-1}$, and 463 cm$^{-1}$ (S-S str.). The shoulder band at 1109 cm$^{-1}$ is suggestive of the presence of propenyl disulfide.

3.4.2. Phytochemicals Identified by GC–MS

GC–MS analysis of garlic peel extracts (Figure S2, Table S1) identified sulfur compounds, such as di-2-propenyl trisulfide or allyl trisulfide (2.1%), thioacetic acid *S*-(tetrahydro-2H-pyran-3-yl) ester (0.6%), disulfide, methyl 2-propenyl (0.2%), diallyl sulfide (0.3%), 5-(3,4-dimethoxyphenyl)-2,3-dihydro-2-thioxo-4-thiazolecarboxaldehyde (0.4%), 2,2-dimethyl-4-ethynyl-tetrahydrothiopyran-4-ol (0.5%), and 2-amino-4-(*p*-aminophenyl)-thiazole (0.3%). In addition to sulfur derivatives, other phytochemicals were also detected: 4-methyl-6-phenyl-pyrimidine (9.5%); d-glycero-l-gluco-heptose (4.6%); glycerin (4.2%); catechol (3%); butanoic acid, 3-oxo-, 1-methylethyl ester (2.6%); hexanoic acid and its esters (2.2%); 9,12-octadecadienoic acid and its esters (2.2%); 5-ethyl-1,3-dioxane-5-methanol (2.2%); octadecenoic acid (2.1%); 2-methoxy-4-vinylphenol (1.7%); lactose (1.6%); 2-hydroxy-gamma-butyrolactone (1.4%); 4-((1E)-3-hydroxy-1-propenyl)-2-methoxyphenol (1.2%); 2,3-dihydro-3,5-dihydroxy-6-methyl-4H-pyran-4-one (1.2%); 5-hydroxymethylfurfural (1.2%); 3-deoxy-*D*-mannoic lactone (1.1%); 2,3,5,6-tetrafluoroanisole (0.9%), and eugenol (0.9%).

## 4. Discussion

*4.1. Phytochemical Profile*

For comparison purposes, phytochemicals reported in essential garlic oil were allyl trisulfide, allyl methyl trisulfide, allyl disulfide, allyl trans-1-propenyl disulfide, allyl methyl sulfide, and 2-vinyl-4H-1,3-dithiine [36]. Allyl trisulfide is the major component of traditional Chinese 'allitridium' medicine. This compound exhibits antimicrobial, anticancer, and antioxidant activities [37,38]. In addition, it has a role as an apoptosis inducer [39], anti-inflammatory agent [40], insecticide [41], antiprotozoal, and antiviral drug [37]. Thioacetic acid *S*-(tetrahydro-2H-pyran-3-yl) ester (or ethanethioic acid, *S*-(tetrahydro-2H-pyran-3-yl) ester) was previously reported to be found in the coffee hull. In regards to 5-(3,4-dimethoxyphenyl)-2,3-dihydro-2-thioxo-4-thiazolecarboxaldehyde and 2-amino-4-(4-aminophenyl)-thiazole, their activity is unknown, but it is expected that, as other thiazoles, they have application in drug development for the treatment of allergies, hypertension, inflammation, schizophrenia, and bacterial and HIV infections [41].

*4.2. Antimicrobial Activity*

Out of all the plant by-products tested in vitro for antifungal activity (Table 1), GPE, followed by OPE, was the most effective against the selected fungal pathogens. To the best of our knowledge, the antifungal activity of extracts made from garlic peels has not been reported before. Nonetheless, extracts made from peeled garlic cloves are known for their antimicrobial activity. The antimicrobial activity of peeled garlic cloves is mainly related to allicin (diallylthiosulfinate), the major sulfur-containing compound of this plant [42], not present in the chemical analysis of GPE. However, the presence of a considerable amount of sulfur compounds (viz. allyl trisulfide, thioacetic acid, methyl 2-propenyl disulfide, diallyl sulfide, etc.) might account for the activity of GPE reported in this study.

in vitro activity of allicin has been previously reported against *B. cinerea* and an oomycete, *Phytophthora infestans*, besides other pathogenic fungi, such as *Alternaria bras-*

*sisicola*, *Alternaria alternata*, *Plectosphaerella cucumerina*, and *Magnaporthe grisea*, and also against the bacteria *Agrobacterium tumefaciens*, *Erwinia carotovora*, *Pseudomonas syringae*, and *Xanthomonas campestris* [42–44]. Regarding OPE antifungal activity, Elsherbiny et al. [45] reported the activity of a methanolic extract inhibiting growth and spore germination of *Fusarium sambucinum*, a causal fungus of dry rot in potato tubers. In addition to garlic and onion peels, and as far as we could find in the literature, the antifungal activity of potato peel extracts has not been reported before. Hence, the activity we report here against the pathogens *D. corticola* and *B. cinerea* (Table 1), which affect very important crops in the Mediterranean basin, suggests the high potential applicability of this plant residue.

In agreement with our study, plant residues from other species presented here have also been shown to have antimicrobial activity. Although activity has been reported against different pathogens and with extracts prepared with different solvents, data are consistent with the hypothesis that they may be a promising source of antifungal compounds or products for agronomical applications. For example, banana peel extract has shown activity against *Fusarium culmorum*, *Rhizoctonia solani* [46], and *Colletotrichum* sp. [47]; *Eucalyptus camaldulensis* bark extract against *Fusarium culmorum* [48]; pine bark and pine needle extract against *Phacidium coniferarum*, *Heterobasidion annosum*, *Nectria ditissima*, and *Alternaria mali* [49,50]; orange peel extract against *B. cinerea*, *A. alternata*, *Monilinia fructicola* and *Aspergillus flavus* [51,52]; lemon peel extracts against *Alternaria flavus* [52]; olive leaf extract against *Penicillium expansum* and *Fusarium proliferatum* [53,54]; and pomegranate peel extract against *Fusarium sambucinum* [55].

Anthracnose of apple is a serious disease caused by numerous species within the *C. acutatum* species complex, which is responsible for up to 50% of yield loss or even more, depending on warmth or moisture conditions [56–58]. In the past, synthetic fungicides such as dithiocarbamate, benomyl, thiabendazole, prochloraz, imazalil, and copper fungicides were used to control anthracnose infection [59]. However, some of them are no longer used because of the introduction of new food safety regulations claiming their harmful effect on human, animal, and environmental health and, in other cases, due to the development of pathogen resistance [15,60,61]. The inhibition of infection progression in inoculated apples (Figures 1 and 2) indicates that GPE holds promise to control fungal infections, at least in fruits after harvest, and can constitute a safe replacement for conventional post-harvest antifungal treatments. As in the present study, the potential of extracts prepared from peeled garlic cloves and compounds extracted from garlic to inhibit fungal infections in plants has been shown in previous works. Daniel et al. [62] demonstrated the potential of garlic aqueous extract in preventing post-harvest decay caused by *B. cinerea* and *P. expansum* in three apple cultivars ('Granny Smith', 'Golden Delicious' and 'Pink Lady'). In mango fruit inoculated with *C. gloesporioides*, the treatment with *Allium longicuspis* ethanolic extract significantly reduced the fruit anthracnose incidence with higher efficiency than the conventional fungicide mancozeb [59]. In addition to phenolic extracts, the essential oil from *Allium sativum* has been demonstrated as effective in reducing the development of decay and anthracnose by inhibiting the mycelial growth and spore germination in strawberry fruit infected by *C. nymphaeae* and banana inoculated with *C. musae* [63,64]. In this work, we demonstrate the antifungal activity of the garlic peel, which contributes to the valorization of the whole plant organ.

### 4.3. Toxicity Mechanism

Due to the pivotal role of ergosterol in fungi plasma membranes and its exclusive presence in this phylogenetic group, this sterol and its biosynthesis became major targets in the development of antifungal products [65,66]. The final part of the pathway of ergosterol biosynthesis, or late pathway, is required for cell viability because deletion of the *ERG* genes is lethal under standard growth conditions. The exceptions are *ERG6*, *ERG2*, *ERG3*, *ERG5*, and *ERG4*, encoding the enzymes catalyzing the last five reactions, presumably due to the similar physicochemical properties that the intermediates accumulated in the corresponding mutants exhibit when compared to ergosterol [66]. Some fungicides applied

for the treatment of plant fungal infections block the ergosterol synthesis, which is the case of morpholines that block the C-14 sterol reduction and C-8 sterol isomerization [65], the azoles that target the enzyme lanosterol 14 α-demethylase [67], and nystatin that binds to ergosterol within the cell membrane to generate pores triggering cell membrane leakage and loss of cytoplasmic content [67]. According to the results with yeast mutants affected in ergosterol biosynthesis (Figure 3), GPE may target fecosterol as this is the only ergosterol precursor accumulated in the mutant that displayed similar sensitivity to that of the parental strain (*erg2*) and that was not accumulated in the mutants that showed higher resistance (*erg6*, *erg5*, and *erg4*). Strikingly, allicin enhances the activity of amphotericin B, an antifungal drug known to target ergosterol, and deletion of *ERG6* in *S. cerevisiae* increases resistance to this combination of compounds [68]. Despite the implication of other mechanisms, this evidence indicates that allicin has a role in the plasma membrane by affecting ergosterol biosynthesis. Similarly to allicin from garlic, other naturally occurring compounds, such as α-bisabolol, a natural phenylpropanoid, have been reported to target the enzyme sterol C-24-methyltransferase, which is responsible for the conversion of zymosterol into fecosterol [69]. The step of conversion of zymosterol into fecosterol is probably an essential one as the target of natural antifungal compounds.

Concerning the extract effect on the cell death process, our results do not indicate the ability of GPE to trigger apoptosis (Figure 4). However, this process should not be excluded since Chin et al. [70] showed that caspofungin fungicide induces cell death without requiring the *YCA1* gene but requiring the pro-apoptotic *AIF1* gene. In addition, allicin has been reported as an inducer of apoptosis via both caspase-independent and caspase-dependent routes. Gruhlke et al. [39] showed that deletion of *AIF1* in yeast promoted resistance to allicin treatment compared to the parental strain. The same authors have hypothesized that at high dosages, allicin promotes necrosis due to high oxidative stress, so the type of death caused by GPE might be a complex process that only further investigations may disclose.

The higher sensitivity of mutants affected in cell wall remodelling observed in this work clearly suggests that GPE targets the fungal cell wall (Figure 5). Similarly to our results, *S. cerevisiae* mutants affected in the cell wall integrity signalling pathway (*pkc1*, *bck1*, and *slt2*) have been demonstrated to be more sensitive to cell wall perturbing agents such as caspofungin, which inhibits β-(1-3)-glucan synthesis one of the main structural polysaccharides of the fungal cell wall [71,72]. Additionally, the deletion of *mpkA*, encoding the *Aspergillus fumigatus* cell wall integrity pathway homolog of the Slt2 protein of *S. cerevisiae*, rendered the cells more sensitive to cell wall-disturbing agents [73]. Therefore, all these data are consistent with a perturbation of the fungal cell wall by GPE.

## 5. Conclusions

The ethanolic extract of garlic peels was found to delay the propagation of *C. acutatum* in apple fruits, suggesting that it may be a promising substitute for synthetic fungicides in the control of fruit diseases. Moreover, our data suggest that the extract may have as a target the integrity of the plasma membrane by affecting the ergosterol biosynthesis and the cell wall. The chemical profile shows the richness of the extract in compounds with diverse and interesting biological properties. It is noteworthy that the variations of sensitivity of the mutant strains shown in this work, although significant, did not have the total reversion of the sensitivity. The most direct explanation for this is the fact that GPE is a complex mixture, probably containing several antifungal components with different cellular targets. The utilization of complex mixtures, such as GPE as biofungicide in agriculture, has the potential benefit of circumventing the selection of resistant strains since the probability of acquisition of resistance against a multitargeting antifungal product would be extremely low. In conclusion, the present study shows that plant by-products can be used to control the growth and propagation of different plant pathogenic fungi, and that, n particular, garlic peels show considerable efficacy, with the potential to be safe for the environment and to avoid selection of resistant strains.

**Supplementary Materials:** The following supporting information can be downloaded at: https: //www.mdpi.com/article/10.3390/horticulturae9020136/s1, Figure S1: ATR-FTIR spectrum of freeze-dried garlic peel extract; Figure S2: GC–MS chromatogram of garlic peel extract and Table S1: Phytochemicals identified in garlic peel extract by GC–MS.

**Author Contributions:** Conceptualization, R.O., A.C., I.C. and G.M.; Methodology, R.O., A.C. and P.M.-R.; Software, A.T.; Validation, R.O., I.C., G.M., A.C. and P.M.-R.; Formal Analysis, A.T., E.S.-H. and J.N.; Investigation, A.T., E.S.-H. and J.N.; Resources, R.O. and P.M.-R.; Data Curation, A.T., E.S.-H. and R.O.; Writing—Original Draft Preparation, A.T.; Writing—Review and Editing, R.O.; Visualization, R.O.; Supervision, R.O., I.C. and G.M.; Project Administration, R.O., I.C. and G.M.; Funding Acquisition, R.O., I.C. and G.M. All authors have read and agreed to the published version of the manuscript.

**Funding:** This work is supported by National Funds by FCT—Portuguese Foundation for Science and Technology (UI/BD/150729/2020)—under the Doctoral Programme "Agricultural Production Chains—from fork to farm".

**Data Availability Statement:** The data presented in this study are available on request from the corresponding author. The data are not publicly available due to their relevance to an ongoing Ph.D. work.

**Acknowledgments:** The authors acknowledge the support of National Funds by FCT—Portuguese Foundation for Science and Technology, under the project UIDB/04033/2020 and the "Contrato-Programa" UIDB/04050/2020 I.P. The authors also acknowledge the financial support by Agri-foodXXI (NORTE-01-0145-FEDER-000041). The authors would also like to acknowledge Pilar Blasco and Pablo Candela from the Technical Research Services of the University of Alicante for conducting the GC–MS analysis. E.S.-H. gratefully acknowledges the financial support of Universidad de Valladolid through the Doctoral Students UVa 2022 Mobility Program.

**Conflicts of Interest:** The authors declare no conflict of interest.

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
