# Peer review of "Antifungal Activity of Plant Waste Extracts against Phytopathogenic Fungi: Allium sativum Peels Extract as a Promising Product Targeting the Fungal Plasma Membrane and Cell Wall"

_horticulturae, doi:10.3390/horticulturae9020136_

Round 1
Reviewer 1 Report
Dear authors,
The paper " A Survey of Antifungal Activity of Plant Waste Extracts against Phytopathogenic Fungi: Allium sativum Peels Extract as a Promising Product Targeting the Fungal Plasma Membrane and Cell Wall" by Ana Teixeira et al. was revised as requested previously and the manuscript fulfils the high standards required for publication in “Horticulturae”.
We analyzed the originality, scientific quality, relevance to the field, presentation and adequacy of the references of the paper.
About the weaknesses of the article:
- Adequacy of the references of the paper: the authors should add more recent references (2022);
- Authors should do an overall review of the references because there are erros (and should standardize references).
This manuscript is acceptable after minor revision.
With kind regards,
Author Response
About the weaknesses of the article:
- Adequacy of the references of the paper: the authors should add more recent references (2022);
- Authors should do an overall review of the references because there are erros (and should standardize references).
This manuscript is acceptable after minor revision.
Authors response: Thank you for your suggestion. New recent references have been included in the manuscript (indicated in blue).
References added:
- Rodrigues, J. A.; Narasimhamurthy, R. K.; Joshi, M. B.; Dsouza, H. S.; Mumbrekar, K. D. Pesticides Exposure-Induced Changes in Brain Metabolome: Implications in the Pathogenesis of Neurodegenerative Disorders. Neurotox. Res. 2022, 40 (5), 1539–1552. https://doi.org/10.1007/s12640-022-00534-2.
- Van den Berg, J.; Greyvenstein, B.; du Plessis, H. Insect Resistance Management Facing African Smallholder Farmers under Climate Change. Opin. Insect Sci. 2022, 50 (100894). https://doi.org/10.1016/j.cois.2022.100894.
- Sánchez-Hernández, E.; Buzón-Durán, L.; Lorenzo-Vidal, B.; Martín-Gil, J.; Martín-Ramos, P. Physicochemical Characterization and Antimicrobial Activity against Erwinia Amylovora, Erwinia Vitivora, and Diplodia Seriata of a Light Purple Hibiscus Syriacus L. Cultivar. Plants 2021, 10 (9). https://doi.org/10.3390/plants10091876.
- Chen, Y.; Fu, D.; Wang, W.; Gleason, M. L.; Zhang, R.; Liang, X.; Sun, G. Diversity of Colletotrichum Species Causing Apple Bitter Rot and Glomerella Leaf Spot in China. Fungi 2022, 8 (7). https://doi.org/10.3390/jof8070740.
In addition, an overall review of the references has been performed, formatting them according to the journal’s guidelines, as suggested.

Reviewer 2 Report
The reviewed work contains the results of a study on the possibility of using extracts obtained from waste products as potential biofungicides. The presented work describes in detail the extraction methodology of the extracts and their chemical composition. Of interest are the results on the potential mechanisms of action especially ergosterol biosynthesis or apoptosis when using an ectract from garlic scales on Colletotrichum acutatum infected apples. The results of the studies discussed here offer the possibility of using these compounds also in protection against other plant pathogens. The results presented indicate a potential fungicidal or fungistatic effect of the tested plant extracts and are therefore valuable information for further practical research. The publication is an interesting contribution indicating the role of ethereal extracts extracted from various plants as important biotic factors limiting the development and abundance of pathogens. This type of work not only has a cognitive aspect, but is also relevant to horticultural practice. The information contained in the publication extends our knowledge of natural substances of plant origin that can be applied in plant protection practice. The research methods adopted are described in detail and do not raise any objections. The results contained in the tables are compiled correctly and statistical tests have been applied correctly. In my opinion, the work can be published in this form.
Author Response
The reviewed work contains the results of a study on the possibility of using extracts obtained from waste products as potential biofungicides. The presented work describes in detail the extraction methodology of the extracts and their chemical composition. Of interest are the results on the potential mechanisms of action especially ergosterol biosynthesis or apoptosis when using an ectract from garlic scales on Colletotrichum acutatum infected apples. The results of the studies discussed here offer the possibility of using these compounds also in protection against other plant pathogens. The results presented indicate a potential fungicidal or fungistatic effect of the tested plant extracts and are therefore valuable information for further practical research. The publication is an interesting contribution indicating the role of ethereal extracts extracted from various plants as important biotic factors limiting the development and abundance of pathogens. This type of work not only has a cognitive aspect, but is also relevant to horticultural practice. The information contained in the publication extends our knowledge of natural substances of plant origin that can be applied in plant protection practice. The research methods adopted are described in detail and do not raise any objections. The results contained in the tables are compiled correctly and statistical tests have been applied correctly. In my opinion, the work can be published in this form.
Authors response: Thank you very much for your positive feedback.

Reviewer 3 Report
In the manuscript “A Survey of Antifungal Activity of Plant Waste Extracts against 2 Phytopathogenic Fungi: Allium sativum Peels Extract as a 3 Promising Product Targeting the Fungal Plasma Membrane 4 and Cell Wall”, authors describe the antifungal activity of different plant waste extracts and study the composition of the most active extract (Allium sativum peel extract) and its possible mechanism of action as an antifungal.
Some comments:
1) Line 22. The antifungal potential of plant by products extracts showed that..
2) Section 3.4. a table would be useful, instead of the enumeration of the different compounds found.
3) Discussion
-line 380-389: authors should first compare the composition of the GPE with the extracts made from peeled garlic cloves, and then compare their antifungal activities.
-Line 448-448. Why do they refer to allicin if it is not present in the GPE? Are there similar compounds in the extract that could act as allicin? I think they should discuss it here (line 487-490).
4) Line 476. Oil refers to the garlic essential oil? Do authors think that the different activities of the peel garlic cloves can be found in the GPE extract? Please revise this paragraph.
I think that the different tests performed on mutants is very interesting and opens the possibility of future studies on the isolation of active compounds and the study of their mechanism of action.
Author Response
In the manuscript “A Survey of Antifungal Activity of Plant Waste Extracts against 2 Phytopathogenic Fungi: Allium sativum Peels Extract as a 3 Promising Product Targeting the Fungal Plasma Membrane 4 and Cell Wall”, authors describe the antifungal activity of different plant waste extracts and study the composition of the most active extract (Allium sativum peel extract) and its possible mechanism of action as an antifungal.
Authors response: We acknowledge and appreciate all the relevant comments, which have been addressed to improve the coherence of the manuscript.
Some comments:
- Line 22. The antifungal potential of plant by products extracts showed that...
Authors response: Corrected accordingly (changes are marked in blue; line 21 in the revised version of the manuscript).
- Section 3.4. a table would be useful, instead of the enumeration of the different compounds found.
Authors response: Thank you for your suggestion. Please kindly note that a table with all the compounds found in the GC-MS analysis is already available in the supplementary material. Only the main ones are mentioned in the main text.
3) Discussion
-line 380-389: authors should first compare the composition of the GPE with the extracts made from peeled garlic cloves, and then compare their antifungal activities.
Authors response: Following the Reviewer’s suggestion, the discussion now starts with a new section on the phytochemical profile, before moving to the antimicrobial activity (lines 380-392 of the revised version of the manuscript).
-Line 448-448. Why do they refer to allicin if it is not present in the GPE? Are there similar compounds in the extract that could act as allicin? I think they should discuss it here (line 487-490).
Authors response: This was already noted at the end of the discussion, but -as suggested by the Reviewer- the information has been relocated and now appears next to the comment on the absence of allicin: “[…] However, the presence of a considerable amount of sulfur compounds (viz. allyl trisulfide, thioacetic acid, methyl 2-propenyl disulfide, diallyl sulphide, etc.) might account for the activity of GPE reported in this study.” (lines 400-402 of the revised version of the manuscript).
- Line 476. Oil refers to the garlic essential oil?
Authors response: Yes, it has been clarified (line 381 of the revised version of the manuscript).
- Do authors think that the different activities of the peel garlic cloves can be found in the GPE extract? Please revise this paragraph.
Authors response: Yes. Please kindly refer to the response to previous comment referring to L448 (please check text in lines 400-402 of the revised version of the manuscript).
I think that the different tests performed on mutants is very interesting and opens the possibility of future studies on the isolation of active compounds and the study of their mechanism of action.

Reviewer 4 Report
The article entitled "A Survey of Antifungal Activity of Plant Waste Extracts against Phytopathogenic Fungi: Allium sativum Peels Extract as a Promising Product Targeting the Fungal Plasma Membrane and Cell Wall" is well written, easy to understand, and should be considered to be published. However, some modifications are suggested to improve the quality of the article.
1. For the title, authors are advised to replace the word "a survey of" with "a screening of" or no need to put these words at all.
2. The image of the line graph of figure 1B, Fig 3, Fig 4, Fig 5. Please put the indicator in the graph what square, round, dash is indicating, to ease the reader's understanding. Even though they have been written in the figure legend, I think it is best to put them in the figure.
3. Typo in subtopic 3.3. (mechanism)
4. The result of GCMS is putative data, is it possible to add the MS/MS data so we can quantify the phytochemicals in GPE?
Author Response
The article entitled "A Survey of Antifungal Activity of Plant Waste Extracts against Phytopathogenic Fungi: Allium sativum Peels Extract as a Promising Product Targeting the Fungal Plasma Membrane and Cell Wall" is well written, easy to understand, and should be considered to be published. However, some modifications are suggested to improve the quality of the article.
Authors response: We greatly appreciate the reviewer’s comments, which have helped us improve the present work.
- For the title, authors are advised to replace the word "a survey of" with "a screening of" or no need to put these words at all.
Authors response: The title has been changed according to the Reviewer’s suggestion.
- The image of the line graph of figure 1B, Fig 3, Fig 4, Fig 5. Please put the indicator in the graph what square, round, dash is indicating, to ease the reader's understanding. Even though they have been written in the figure legend, I think it is best to put them in the figure.
Authors response: The suggested alterations were done.
- Typo in subtopic 3.3. (mechanism).
Authors response: Corrected accordingly.
- The result of GCMS is putative data, is it possible to add the MS/MS data so we can quantify the phytochemicals in GPE?
Authors response: Generally, tandem MS/MS instruments are used for target analysis only and not for the routine analysis of unknowns. We are aware that MS/MS instruments can also be used as a standard MS—this is required during method development of MS/MS techniques—however, caution should be applied as the mass spectrum produced may be of a lower quality and affect the identification, particularly for scanning instruments. Hence, we do not share the Reviewer’s view on the reported GC-MS data (assuming that putative has been used in the sense of indicating that it may not be certain). Taking into consideration that GC-MS (not GC-MS/MS) is widely used in the characterization of extracts and essential oils and is generally regarded as a ‘gold-standard’ (many examples may be found, for instance, in Plants latest issue); that the use of multiple stages (mass analyzers) within the mass spectrometer to increase the sensitivity of the analyte by reducing the background from the coeluting matrix peaks is not required in this case; and that the requested information on the quantification of phytochemicals in GPE is already provided in the ‘Area (%)’ column (which indicates the amount of the specific analyte that is present), no additional data has been included in the manuscript.
